# Augmented Reality in Neurosurgery: A New Paradigm for Training

**DOI:** 10.3390/medicina59101721

**Published:** 2023-09-26

**Authors:** Grace Hey, Michael Guyot, Ashley Carter, Brandon Lucke-Wold

**Affiliations:** 1College of Medicine, University of Florida, Gainesville, FL 32610, USA; 2Eastern Virginia Medical School, Norfolk, VA 23507, USA; 3Department of Neurosurgery, University of Florida, Gainesville, FL 32610, USA

**Keywords:** augmented reality, machine learning, neurosurgery, cranial neurosurgery, spine surgery, neuro-oncology, neurovascular surgery, neurosurgical training

## Abstract

Augmented reality (AR) involves the overlay of computer-generated images onto the user’s real-world visual field to modify or enhance the user’s visual experience. With respect to neurosurgery, AR integrates preoperative and intraoperative imaging data to create an enriched surgical experience that has been shown to improve surgical planning, refine neuronavigation, and reduce operation time. In addition, AR has the potential to serve as a valuable training tool for neurosurgeons in a way that minimizes patient risk while facilitating comprehensive training opportunities. The increased use of AR in neurosurgery over the past decade has led to innovative research endeavors aiming to develop novel, more efficient AR systems while also improving and refining present ones. In this review, we provide a concise overview of AR, detail current and emerging uses of AR in neurosurgery and neurosurgical training, discuss the limitations of AR, and provide future research directions. Following the guidelines of the Preferred Reporting Items for Systematic Reviews and Meta-Analyses (PRISMA), 386 articles were initially identified. Two independent reviewers (GH and AC) assessed article eligibility for inclusion, and 31 articles are included in this review. The literature search included original (retrospective and prospective) articles and case reports published in English between 2013 and 2023. AR assistance has shown promise within neuro-oncology, spinal neurosurgery, neurovascular surgery, skull-base surgery, and pediatric neurosurgery. Intraoperative use of AR was found to primarily assist with surgical planning and neuronavigation. Similarly, AR assistance for neurosurgical training focused primarily on surgical planning and neuronavigation. However, studies included in this review utilize small sample sizes and remain largely in the preliminary phase. Thus, future research must be conducted to further refine AR systems before widespread intraoperative and educational use.

## 1. Introduction

AR technology involves the superimposition of computer-generated images onto the user’s real-world visual field to modify or enhance the user’s visual experience. In medicine, AR has been utilized in various surgical subspecialties, including ophthalmology, general surgery, orthopedic surgery, urology, and oral and maxillofacial surgery for its ability to integrate computer-generated images with intraoperative visualization of the surgical field [1]. The field of neurosurgery in particular has been at the forefront of image-guided surgery since the early 1990s, as surgeons must demonstrate high intraoperative accuracy and precision to navigate microscopic targets in a way that maximizes patient outcomes while also preserving neurologic functioning. Conventional neuronavigation and neuroimaging technologies involve the use of two-dimensional images to guide surgical planning and neuronavigation. However, integration of preoperative and/or intraoperative magnetic resonance imaging (MRI), computerized tomography (CT), angiography, and tractography images into the real-world surgical environment provides neurosurgeons with an enriched, three-dimensional (3D), semi-immersive experience designed to improve surgical planning, accuracy, and precision. Surgical assistance with AR has been shown to improve neuronavigation, enhance surgical planning, and reduce procedure duration in both spinal and cranial neurosurgery. Accordingly, the use of AR in neurosurgery has significantly increased over the last decade [1]. Emerging pre-clinical studies regarding the use of AR in neurosurgery aim to improve neuronavigation, reduce operation times, and improve patient outcomes.

It should be emphasized how the innate properties of AR facilitate enhanced neurosurgical training. In both cranial and spinal neurosurgery, AR systems are largely designed to project 3D images of cerebral or spinal anatomy onto realistic patient models to aid in neuronavigation and surgical planning. These superimposed images are either injected into the operative microscope or worn as a head-mounted display, allowing for easy and efficient use that does not impact on surgical technique or obstruct the view of the surgical field. As such, AR systems provide the opportunity for refined neuronavigation, assistance with identification of microscopic anatomic structures, improved surgical planning, and reduced operation times. These elements of AR effectively provide neurosurgeons with training materials that can improve surgical planning and surgical technique while simultaneously minimizing patient risk.

It is important to note that many forms of AR utilized for intraoperative assistance and training remain largely in the emerging or preliminary phases, highlighting the need for continued research efforts before widespread clinical use. Nonetheless, AR has the potential to significantly improve and revolutionize the field of neurosurgery. Herein we discuss the current uses of AR in cranial and spinal neurosurgery, emerging pre-clinical studies, and future research indications.

## 2. Overview of AR

AR is a computer-based technology allowing users to overlay visual information in their field of view by combining real and virtual objects in the real environment, running in real-time, and connecting real and virtual objects [2]. All three conditions are satisfied to create an effective AR system. Unlike Virtual Reality (VR) systems, AR platforms are not fully immersive, but rather a mixture of real and virtual environments. Hardware with the capability to run Augmented Reality Applications (ARAs) include but are not limited to headsets, smart glasses, and mobile devices [3]. These devices use sensors, including cameras, accelerometers, global positioning systems, and solid-state compasses to capture information about the real-world surrounding environment. The processing components of the hardware then transform this information and overlay the appropriate output information to create an augmented view of the user’s visual field [2]. Figure 1 presents a brief overview of how AR has been integrated into the field of neurosurgery.

Up to this point, AR platforms have been used in a variety of different industries, including the entertainment [4], automotive [5], and healthcare industries. Within healthcare, specialties that require technical skills and dexterity have been more likely to adopt AR systems. Most notably, ARAs have been implemented in dentistry [6], oral and maxillofacial surgery [7], ophthalmology [8], orthopedics, and plastic surgery [9]. In addition, AR platforms have shown to be effective in the treatment of psychological disorders such as phobias and anxiety [2]. In medical education and training, ARAs have been incorporated into teaching anatomy, classroom study aids, and clinical training simulators [8]. For example, due to the high cost and limited availability of cadavers, some medical schools are turning to ARAs to provide foundational anatomic education to medical students. Within the realm of surgery, AR allows students to practice surgical techniques in a highly realistic environment without the need for a physical patient, reducing potential risks [3]. With this reduced risk, students are able to step outside of their comfort zone and attempt more complex surgeries without compromising patient safety. Students can practice using the simulation until they feel more capable in their abilities, and once their skills have progressed, they can begin operating on physical patients.

While AR systems generate a virtual landscape, they are able to provide haptic feedback to trainees, allowing them to develop their skills [9]. AR additionally allows for remote learning and teaching. This process involves an international surgeon using an augmented reality proctoring system where a remote surgeon will assist in the marking of various anatomical landmarks to reduce the risk of complications. One study found that this platform was a safe, reliable, and effective way to remotely conduct cleft deformity repair surgery [10].

Given the increasing prevalence of AR use within various surgical subspecialties, there exists a need to highlight recent applications of AR for intraoperative assistance during cranial and spinal neurosurgery. Furthermore, the potential for AR to be used as a teaching tool demonstrates an additional need to summarize recent advances in AR systems specifically for neurosurgery training programs. This systematic review aims to summarize recent applications of AR systems for intraoperative assistance in both cranial and spinal neurosurgery, as well as recent applications of AR assistance for neurosurgical training.

## 3. Methods

A systematic literature search was performed in June 2023 using PubMed and following the guidelines of PRISMA. Customized queries included the following keywords with AND/OR operators entered into the search engine: AR, augmented reality, neurosurgery, cranial neurosurgery, spinal neurosurgery, and neurosurgical training. The search was limited to full-text original retrospective or prospective studies, or case reports published in English after the year 2013. The search was limited to studies published between 2013 and 2023 in attempts to provide the most up-to-date information, which we defined as studies published within the past 10 years. Article screening and inclusion eligibility was conducted independently by two different reviewers (GH and AC). Studies that evaluated the safety and efficacy of AR systems used for intraoperative assistance during cranial or spinal neurosurgery were included. Similarly, studies that evaluated the feasibility and efficacy of AR integration into education with respect to neurosurgical training were included. A total of 386 articles were initially identified based on our search strategy. After removing obviously irrelevant articles based on title alone, a total of 83 abstracts that fit the inclusion criteria were retrieved and screened. After screening abstracts and removing duplicate articles, 31 total articles published between 2013 and 2023 were identified and included in this review. Original (retrospective and prospective) research articles as well as case reports published in English were included in this systematic review (Figure 2).

## 4. Results

### 4.1. Current Use of AR in the Operating Room for Cranial Neurosurgery

AR is primarily used in cranial neurosurgery to improve technical skills and assist in surgical planning [11], though it can additionally be used to locate brain structures and guide surgical resections [12]. For example, image-guided neurosurgery allows for more accurate surgical planning, intraoperative navigation, and trajectory planning in stereotactic neurosurgery [13,14]. As a more specific example, an understanding of white-matter tract anatomy is critical in cranial neurosurgery. An AR model built with photogrammetry utilizing cadaveric brain specimens has been used to produce high-resolution models of white-matter tracts. These models can then be shifted and rotated on different planes, visualized from multiple angles and magnifications, and projected on any real surface [15]. This form of AR has the potential to train and guide physicians to better understand white-matter tract anatomy and intraoperative location. AR has additionally been used to assist in the surgical navigation of intraventricular pathologies [16]. Specifically, an AR system constructed with Scopis NovaPlan navigation software has been used to plan and project surgical regions of interest and surgical trajectories onto the endoscopic video screen during surgery [16]. A clinical study utilized this AR modality to successfully assist with cyst fenestrations, biopsies, endoscopic third ventriculostomies, stent placements, and shunt implantations, demonstrating that AR has the ability to improve the safety and efficacy of intraventricular neuroendoscopic surgery [16].

Presently, AR is most commonly used in neurosurgery for neurovascular pathologies [17], followed by neuro-oncological pathologies and non-vascular and non-neoplastic lesions [12]. AR has been utilized in cerebrovascular neurosurgery to provide assistance with the treatment of arteriovenous malformations (AVMs) [18], cerebral aneurysms [19], and extracranial-intracranial bypass revascularization surgery [20]. Intracranial AVMs are lesions containing pathological connections between arteries and veins in the brain. AVM lesions are often highly complex, and subsequent dysregulation of the homeostatic capillary architecture can result in hemorrhagic cerebrovascular accidents if left untreated. AR has the potential to guide surgeons in navigating the complex AVM vasculature to treat patients without causing damage to the normal surrounding vasculature. In a case report detailing a patient with a ruptured right parietal Spetzler-Martin grade I AVM secondary to a mechanical fall, AR was used to project a hologram of the angioarchitecture on the surface of the cortex, providing in the object recognition an image of the angiographic anatomy during surgery [18]. Six months after the procedure, the AVM was completely obliterated and the patient experienced complete relief of symptoms, demonstrating that AR-assisted microsurgery can improve the safety and efficacy of AVM resection [18]. Surgical intervention for the treatment of cerebral aneurysms is similar to that of AVMs in the sense that surgery must be conducted in a highly precise manner to avoid damaging the surrounding vasculature. To this end, Cabrilo et al. [19] combined preoperative 3D angio-magnetic resonance imaging, angio-computed tomography, and 3D digital subtraction angiography images to generate virtual segmentations of patients’ vessels, aneurysms, aneurysm necks, skulls, and heads. These images were then projected into the eyepiece of the operating microscope during surgery, resulting in improved neuronavigation and optimized aneurysm clipping [19]. AR has been similarly designed to assist with extracranial-intracranial bypass revascularization surgery by combining preoperative images to visualize cerebral architecture [20].

In addition to cerebrovascular surgery, AR has a variety of applications within the subfield of neuro-oncology. The clinical management of many cerebral malignancies typically involves surgical resection, where surgeons aim to maximize the total amount of tumor resected in a way that improves overall patient survival while also preserving neurologic functioning. AR has the ability to improve this process by aiding in neuronavigation, ultimately resulting in more complete tumor resections [21]. To improve surgical planning and overall preciseness, Ivan et al. [22] designed a head-mounted display able to identify and display circumferential tumor borders on HoloLens AR glasses worn by surgeons in the operating room. The authors were able to demonstrate the feasibility of tumor border identification for incision planning using this form of AR, though future studies are needed to further refine this practice [22]. Furthermore, AR has the ability to enhance surgical planning for intradural tumors through the use of intraoperative navigation-integrated projections onto the surface of the skull [23]. In this way, the process of opening the skull to remove skull-base lesions is refined, providing a less invasive operative experience for patients, which ultimately manifests in improved patient outcomes [23]. In the case of transsphenoidal surgery, which involves surgical navigation through the nasal passages and sphenoid sinus, AR is able to improve landmark identification and intraoperative navigation [24]. For example, AR has been used to assist in transsphenoidal pituitary adenoma resection surgery by projecting preoperative CT images onto the operating microscope [25]. AR assistance during surgery was found to significantly improve surgical accuracy (target registration error, 0.76 ± 0.33 mm) as compared to surgical intervention without AR assistance (target registration error ± 1.02 mm) [25]. The creation of an AR system from preoperative radiographic data superimposed onto the surgical field and video monitor has led to similar beneficial effects of improved neuronavigation in patients receiving endoscopic transsphenoidal skull-base surgery for the treatment of sellar and parasellar tumors [26]. These results ultimately demonstrate the ability of AR to improve surgeon accuracy and patient safety.

### 4.2. Current Uses of AR as a Training Tool for Cranial Neurosurgery

In addition to offering assistance during surgery, AR has emerged as a valuable tool for neurosurgical training. This is because AR provides surgeons with training materials that minimize patient risk. For example, ventricular puncture is a common procedure performed in neurosurgical practice that involves creating an opening in the skull to provide ventricular access to treat ventricular pathologies. AR designed to project 3D images of cerebral anatomy over a 3D-printed patient model was able to improve surgical trajectory accuracy (n = 48 neurosurgeons), demonstrating that AR can be a safe alternative for neurosurgical training [27]. External ventricular drains and needle biopsies are two additional common procedures in neurosurgical practice that require high precision. Skyrman et al. [28] evaluated the accuracy (deviation from the target or intended path) and efficiency (insertion time) of an AR navigation system in both procedures through the use of intraoperative CT images and integrated video tracking of brain structures [28]. Participants were asked to reach a 2 mm spherical biopsy target on a 3D-printed skull phantom with air-filled ventricles and realistic gelatinous brain structures [28]. AR-guided needle biopsy insertions with this strategy had an accuracy of 0.8 mm ± 0.43 mm, and the average insertion time was 149 s [28]. Similarly, AR-guided external ventricular drain accuracy was 2.9 mm ± 0.8 mm at the tip with a 0.7° ± 0.5° angular deviation from the planned track, and the average insertion time was 188 s [28]. This demonstrates that AR has the ability to aid in training for external ventricular drain placement and needle biopsy insertion, with potential for future integration into clinical practice [28]. To improve the overall accuracy, efficiency, and intuitiveness of external ventricular drain placement, Chiou et al. [29] developed an AR system designed to superimpose the surgical target position, scalpel entry point, and scalpel direction onto the head of the patient. These data were then displayed on a tablet, allowing for efficient identification of the surgical target and preferred entry point [29]. This particular form of AR provided highly accurate surgical visualization and improved surgical efficiency in laboratory and hospital based trials, highlighting its efficacy for both training and implementation into neurosurgical practice [29].

AR has additionally been used as a valuable tool for training surgeons how to locate, navigate, and treat malignant lesions. A study by Montemurro et al. [30] investigated the potential of AR assistance in craniotomy trajectory planning for the surgical resection of meningiomas by projecting preoperative MRI images from a head-mounted display onto a patient-specific mannequin. Here, the surgeon was able to create a bone flap and trace the surgical trajectory with an error of less than ±1 mm, ultimately demonstrating that head-mounted AR systems have the potential to assist in presurgical planning and surgical trajectory training [30]. As aforementioned, tumor resection completeness is a key part of clinical management for many malignant brain lesions. AR has the ability to assist with training surgeons to plan and navigate tumor resections through the use of HoloLens software. This form of AR utilizes built-in infrared tracking to project patient-specific anatomical images onto a head-mounted display [31]. In a phantom study utilizing HoloLens AR to determine tumor borders for optimal surgical planning, registration errors remained below 2.0 mm, tumor delineation was deemed superior in 65% of cases, and the overall surgical planning time was significantly reduced [31]. This study employed a total of 12 surgeons with varying degrees of experience, highlighting the potential HoloLens AR has for improving neurosurgical training [31]. HoloLens AR has additionally been used for trajectory planning for craniofacial surgery in pediatric patients [32]. Specifically, AR guidance with the HoloLens head-mounted display allows surgeons to successfully trace the surgical trajectory with an accuracy level of ±1.5 mm [32]. Though this was an in vitro study, the results suggest this form of AR has promise for both training and clinical use [32]. It is important to note that the intraoperative use of AR for neuro-oncological surgery primarily focuses on surgical planning and intraoperative neuronavigation. However, there are other important aspects of cranial malignancy pathology, such as peritumor vessels, vascular pedicle dysregulation, and dentification of the peritumoral brain zone, where the use of AR for treatment has not been explored in the literature. Thus, future studies should be conducted to investigate how AR can be used to study peritumoral vessels, the vascular pedicle in the case of metastases, and identification of the peritumoral brain zone to ultimately expand the use of AR for intraoperative treatment of various malignancies.

### 4.3. Current Uses of AR in the Operating Room for Spinal Neurosurgery

AR in spine surgery is currently used for spinal fusions, vertebroplasty, kyphoplasty, neuro-oncology, and spinal deformity correction [33,34]. One common application of AR in spinal neurosurgery is AR-assisted pedicle screw placement, which involves connecting vertebrae in the spine with screws to preserve motion segments and stabilize the spine to promote healing after surgery [35]. Though highly accurate, conventional pedicle screw placement is performed in a freehand manner and relies on correct visual identification of anatomical landmarks to determine the best surgical trajectory. Misplaced screws can cause significant morbidity; a risk that may be minimized with intraoperative AR assistance. Liu et al. [36] specifically tested this idea by developing a head-mounted display designed to acquire and register 3D images obtained from O-arm and C-arm neuronavigation systems during thoracic, lumbar, and/or sacral pedicle screw placement. This form of AR was found to provide surgical accuracy of 98.0%, demonstrating that AR has the ability to significantly improve pedicle screw placement accuracy [36]. A later study by Terander et al. [37] optimized this system by mounting four optical cameras into the frame of the C-arm. Once the surgeon inputs the screw dimensions and optimal screw path, the C-arm uses the cameras to produce 3D CT scans that project augmented images containing the optimal surgical path onto the surgical field [37]. This form of AR was found to yield an accuracy of 94.1% with no device-related adverse events, highlighting the efficacy and safety of AR-assisted surgical navigation for pedicle screw placement [37]. More modern AR systems have the ability to wirelessly display intraoperative 3D CT images onto a headset with transparent near-eye displays, creating a 3D “see-through” effect in addition to standard 2D neuronavigation images. [38]. This novel AR system was found to significantly reduce surgical operation time, improve neuronavigation, and improve patient outcomes with no adverse events in 155 cases [38].

Spinal deformity correction surgeries are complex procedures that require intensive preoperative planning and highly precise and accurate surgical technique. AR assistance for spinal osteotomy was first reported by Kosterhon et al. [39]. In this case report, 3D virtual resection planes were created from preoperative CT images of a 56-year-old female receiving an osteotomy for a congenital wedge-shaped hemivertebra between T12 and L1 [39]. These 3D resection planes were then injected into the surgical microscope, ultimately allowing the surgeon to better visualize the surgical target [39]. Although this novel AR system was only used in a single patient, it demonstrated the potential of AR assistance for spinal osteotomies. Similarly to osteotomies, percutaneous vertebroplasty can be used to correct some spinal deformities. Percutaneous vertebroplasty involves surgical injection of bone cement into the vertebral body to provide structural support following vertebral fracture [40]. Although this is considered to be a minimally invasive procedure, needle placement and surgical planning must be executed in a highly accurate and precise manner to achieve lasting functional benefits. Surgical assistance with AR has the ability to enhance percutaneous vertebroplasty through enhanced neuronavigation, improved pre-surgical planning, and decreased operation time. For example, Hu et al. [41] constructed an AR system designed to superimpose 3D images from pre-operative CT scans onto the surgical field. This system was found to decrease radiation exposure, reduce operation times, and improve bone entry point accuracy [41]. Neuronavigation and surgical accuracy can further be refined by visualizing a needle trajectory in 3D space and overlaying that image onto the surgical field through use of a head-mounted display [42]. This AR system, known as VIPAR, has demonstrated improved needle projection planning in forty phantom trials and five clinical trials with no patient complications [42,43].

AR can be used to assist with surgical resection of malignant lesions present in the spine. Identification of malignant lesions within the spinal column is accomplished via surgical biopsy, and clinical management often involves surgical resection. As with cranial neurosurgery, surgical resection of malignant lesions within the spine must be conducted in a careful manner to ensure vital structures within the spinal column are not disturbed, as this could result in significant neurologic deficits. AR has the ability to enhance spinal tumor resection procedures by aiding neurosurgeons in tumor-outline visualization through the projection of preoperative CT and MRI images onto a heads-up display within the operating microscope [44]. En bloc tumor resections represent a specific resection procedure type and aim to remove tumors in a single, intact piece, fully encased by healthy tissue. Although this is currently the most effective way to remove tumors within the spine, en bloc tumor resections have the potential to cause significant morbidity due to the large areas of exposure, the complex nature of the surgery, and the tumor location near functionally and anatomically important structures [45]. To enhance neuronavigation for surgical resections of sacral and retrorectal lesions, Tigchelaar et al. [45] developed a form of AR using the SynchAR visualization platform, designing it to display augmented images of spinal anatomy within the operative microscope. Intraoperative use of this AR system resulted in enhanced neuronavigation, smaller surgical incisions, and reductions in bony resection extent, with no known patient complications [45]. This study was the first of its kind to use AR assistance for en bloc resections of spinal column lesions, highlighting the overall efficacy and safety of this novel AR system. Later studies have investigated the use of this AR form for an en bloc wide marginal resection of an L1 chordoma and have demonstrated similar benefits of improved neuronavigation with minimal patient complications [46]. AR can further be used to enhance resection of intradural extramedullary tumors. Intradural extramedullary tumors are generally benign neoplasms of the spinal canal that represent approximately 15% of all central nervous system tumors [47]. AR systems designed to project preoperative CT and MRI images into the surgical microscope have the ability to assist with tumor localization and reduce operation times, highlighting the feasibility of this AR form [48].

### 4.4. Current Uses of AR as a Training Tool for Spinal Neurosurgery

AR has the ability to facilitate and enhance spine surgery training while also offering assistance in the operating room. As aforementioned, pedicle screw placement is commonly used to stabilize the spine during surgery and promote functional recovery post-operatively. Pedicle screw placement can be utilized in spinal fusion operations, where bones in the spine are permanently fused together to correct abnormal spine morphology, repair broken bones, or stabilize the spine to prevent movement. Following pedicle screw placement, rods are used to connect the pedicle screws, effectively preventing additional movement while simultaneously supporting the fusion of the target vertebrae [49]. Rod manipulation is a surgical technique required to appropriately place hardware within the vertebral column. If manipulated incorrectly, rod complications can lead to screw loosening, screw removal, increased operation times, and worse patient outcomes. To improve neuronavigation during rod manipulation, Wanivenhaus et al. [50] developed a form of AR using HoloLens software. After placing a spatial anchor near the surgical target in a realistic model of a human spine, the HoloLens camera is able to acquire images and project them onto the head-mounted display worn by the surgeon [50]. Use of the HoloLens system was found to significantly reduce rod insertion times and improve surgical accuracy, ultimately demonstrating that this form of AR can be used as a valuable training tool on realistic mannequins before training on live patients [50]. To further improve and refine the navigation process required for rod bending and implantation, Von Atzigen et al. [51] created a form of AR designed to provide surgeons with a step-by-step guide for rod bending. Here, intraoperative video stream data acquired from the HoloLens software was used to train a stereo neuronal network that would provide the surgeon with digitized screw head positions, optimal rod shape, and optimal bending parameters [51]. This AR form was tested in human cadavers and was found to significantly decrease rod bending maneuvers when compared to the conventional free-hand approach [51]. Enabling trainees to practice neuronavigation and surgical procedures enhanced with AR on cadavers allows for step-by-step training in a highly realistic environment without risking patient safety. Although future studies should be conducted to further refine this training modality, the results of this study demonstrate the potential for enhanced rod bending and implantation following pedicle screw placement.

Similarly to percutaneous vertebroplasty, percutaneous kyphoplasty involves the insertion and inflation of a balloon into the surgical space, and is designed to create a cavity before injection of bone cement [52]. Both are effective surgical interventions for the treatment of vertebral compression fractures, though recent evidence suggests percutaneous kyphoplasty may be favored for improved vertebral height restoration and reduced post-operative pain [53]. AR has been shown to assist with percutaneous kyphoplasty through use of the HoloLens system [54]. Specifically, projection of preoperative images onto a lumbar spine phantom while wearing HoloLens glasses was found to reduce procedure times and improve visual intraoperative guidance [54]. If trainees are allowed to use HoloLens glasses on realistic models of the spine, surgical technique and neuronavigation can be improved. AR has additionally been utilized to facilitate lumbar facet joint injection trainings. Lumbar facet joint blocks involve injection of a local anesthetic into the joints localized alongside each vertebra to diagnose and manage facet joint pain. HoloLens AR developed from CT images of the spine has an injection accuracy of 97.5% compared to injections guided with CT images alone, which demonstrated an accuracy of 100% in a spine phantom [55]. Although guidance with AR was not perfect in this case, the results demonstrate that AR has the potential to assist with identification of anatomic structures in neurosurgical training.

## 5. Emerging Studies Investigating the Role of AR in Neurosurgery

As demonstrated in our review of the recent literature, AR is rapidly evolving for use in neurosurgical procedures and neurosurgical training. However, despite technological advances in AR software and supporting hardware modalities, few randomized control trials have been conducted to evaluate patient outcomes in large cohorts, including outcomes related to postoperative complications and cost-effectiveness [56]. A large number of studies included in this review tested the use of AR in relatively small sample sizes, indicating that the current body of research regarding the applications of AR in neurosurgery is largely focused on improving and refining AR software and hardware before clinical use in a larger group of patients.

There currently exist two available registered clinical trials investigating the use of AR in neurosurgery (Table 1). One trial at the University of Pennsylvania (NCT03921385) was designed to investigate the safety and efficacy of AR assistance with holographic images projected onto the surgical field for neuronavigation in 32 patients receiving cranial or spinal surgery. This clinical trial was completed on 23 February 2023, though no results have been published in the literature at this time. Another study at Balgrist University Hospital (NCT04610411) is currently recruiting participants for a study aiming to investigate the accuracy of implant navigation using AR glasses to improve pedicle screw and rod implantation (Table 2).

Similarly, many reports included in this review detailing the applications of AR for neurosurgical training remain largely in the early stages of development. More specifically, many forms of AR designed to assist with neurosurgical instruction and training currently reported in the literature focus on teaching scenarios in small sample sizes. AR systems designed to aid in surgical instruction in a highly realistic manner and with the ability to account for individual patient differences take a significant amount of time to develop and refine. The studies detailed in this systematic review ultimately suggest that the use of AR in neurosurgical training remains largely in the developmental stages. Once its use been refined, it would be valuable to conduct studies that evaluate the feasibility and efficacy of AR for assistance with neurosurgical training in a large group of neurosurgeons, each with varying skillsets and years of experience. In this way, it would be possible to better understand how AR can be utilized to train a diverse group of trainees to further refine AR-based training modalities.

## 6. Conclusions and Future Research Directions

It is well established throughout the literature that AR has the potential to significantly improve neurosurgical practice through enhanced neuronavigation, improved surgical planning, and reduced operation times. The efficiency and enhanced operative experience that AR assistance offers neurosurgeons ultimately manifests in improved patient outcomes and cost-effective options for surgeons [57]. In addition, AR has the ability to optimize neurosurgical training for providers with varying degrees of experience in a way that provides a realistic operative experiences while minimizing patient risk. This property of AR allows for providers to refine and master complex surgical techniques and intraoperative neuronavigation for lesions that appear to be difficult to reach or are located near vital brain structures.

It is important to note that the optimal utilization of AR exploits its ability to enhance surgical and training experiences, rather than relying exclusively on it for training, surgical planning, and neuronavigation. This is because despite the apparent benefits of AR-assisted neurosurgery, there are several limitations and challenges preventing translation to clinical practice. Furthermore, most of the studies currently reported in the literature utilize small sample sizes, which may not be fully representative of all patients receiving AR-assisted neurosurgery. For example, ensuring high accuracy and precision in the registration of virtual objects onto the patient’s cerebral anatomy is crucial. Introduction of AR into the operating room has the potential to distort surgical accuracy and precision due to impairments in surgeon depth-perception, visual and tactile asynchrony between the AR and the surgeon, tissue deformation, and technical inaccuracy in AR tracking systems. It is possible for structures in the brain to slightly shift once the skull is open, which can result in surgical errors if not accounted for. Future research directions should look at how AR can predict and account for these errors to further refine surgical practice. In addition, the hardware required for intraoperative AR use, including tracking devices and head-mounted displays, must be designed in a way that is ergonomic and capable of delivering high-quality information in real time. Hardware demands have the potential to serve as a barrier to clinical use, and future studies should be conducted to determine the optimal materials and setup needed to support AR use during surgery. AR systems often utilize a diverse mix of preoperative and real-time intraoperative imaging data. As such, it is possible for errors to arise when integrating different data forms, which may impose on surgical planning and neuronavigation. Similarly, seamless integration of AR into the surgical workflow is essential to ensuring surgical efficiency and optimal surgeon focus. Introduction of novel AR technology into the operating room has the potential to create technological barriers during surgery, which increase surgery time, distract the surgeons, and may negatively impact patient outcomes. To combat these challenges, neurosurgeons and the entire care-providing team must undergo significant training that covers learning new techniques, adapting to the augmented environment, and understanding the limitations of AR to become fully proficient with AR technology. Thus, it is important that future studies continue to focus on refining AR systems to make them more user-friendly to avoid complications during surgery, and on developing education efforts.

The overall cost of AR technology can additionally serve as a barrier to clinical implementation, especially in smaller hospitals or clinics that may not benefit from significant research grants or large private donors. Adequate funding is necessary to maintain AR systems, including repairs, software development, hardware maintenance, and training. AR systems are constantly evolving, and significant monetary input is necessary to keep up with software and hardware advances. The use of AR in the intraoperative environment requires the presence of additional staff, such as engineers or software developers, to attend to technological issues that may arise during surgery. Without the presence of these staff members, surgical times can be delayed, AR systems can experience significant damage, and patient safety can be compromised. It should not be surprising to note that the cost of AR systems ultimately impacts their availability. The use of AR in neurosurgery remains largely in the preclinical phases, highlighting the need for continued research efforts before widespread clinical adoption. Many institutions currently utilizing AR in neurosurgery are conducting laboratory-based experiments and human clinical trials. Thus, institutions must possess sufficient funding to support these research efforts to ultimately refine AR systems before clinical implementation. Furthermore, if institutions lack the necessary funding to support the use of AR during surgery, it is possible that the AR software or hardware could become faulty, resulting in a significant financial loss. More importantly, the use of damaged AR systems or AR systems that are not properly maintained can pose a significant risk to patient safety. Accordingly, if institutions cannot afford to properly maintain AR systems and keep up with the advances in technology, they cannot utilize such systems during surgery without creating a significant risk of financial loss, equipment damage, patient morbidity, and patient mortality; this limits the overall viability of AR systems. This is a challenge that must be addressed as AR continues to develop and evolve to ensure all patients have access to the highest-quality neurosurgical care.

Additional limitations preventing implementation of AR in neurosurgical training exist. For example, a major limitation of AR in neurosurgery is its inability to provide a fully realistic experience for trainees. This is of utmost importance, because although AR can be used to enhance neurosurgical training, neurosurgeons ultimately practice medicine on real patients. Accordingly, AR simulations for training must be designed in a way that creates a realistic visual and tactile experience to maximize surgical education. Similarly, it can be difficult to develop assessment tools and objective metrics that provide trainees with meaningful feedback. Future studies should investigate how to develop more realistic AR simulations for neurosurgical trainees to ensure surgical proficiency. Integration of AR training into the existing neurosurgical curriculum additionally poses a significant challenge. Careful planning and coordination with training programs and educational institutions is necessary to incorporate AR modules into training without disrupting or reducing the quality of pre-existing training modules. Furthermore, developing and providing access to high-quality AR training platforms specifically tailored for neurosurgery can be a challenge, especially for smaller or newer training programs that may lack substantial financial resources. Thus, it is imperative to address these challenges to ensure affordable and accessible AR training platforms to trainees. Finally, neurosurgical training programs must account for and accommodate the diverse range of skill levels and needs of training neurosurgeons. It can be difficult for AR to adapt to individual needs, which means that future research must be conducted to personalize and further refine the neurosurgical training process.

In sum, AR has the potential to revolutionize neurosurgical training and practice. At this point in time, physician oversight is still necessary during neurosurgical training to ensure trainees are receiving the highest-quality training to provide the highest-quality care. More studies need to be conducted to further validate and verify the accuracy and reliability of current AR systems as well as to facilitate expansion to additional neurosurgical procedures.

## Figures and Tables

**Figure 1 medicina-59-01721-f001:**
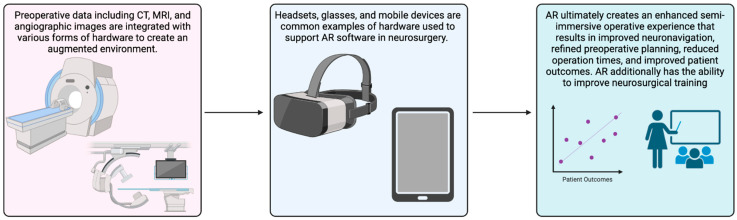
Brief visual overview of how AR has been integrated into the field of neurosurgery.

**Figure 2 medicina-59-01721-f002:**
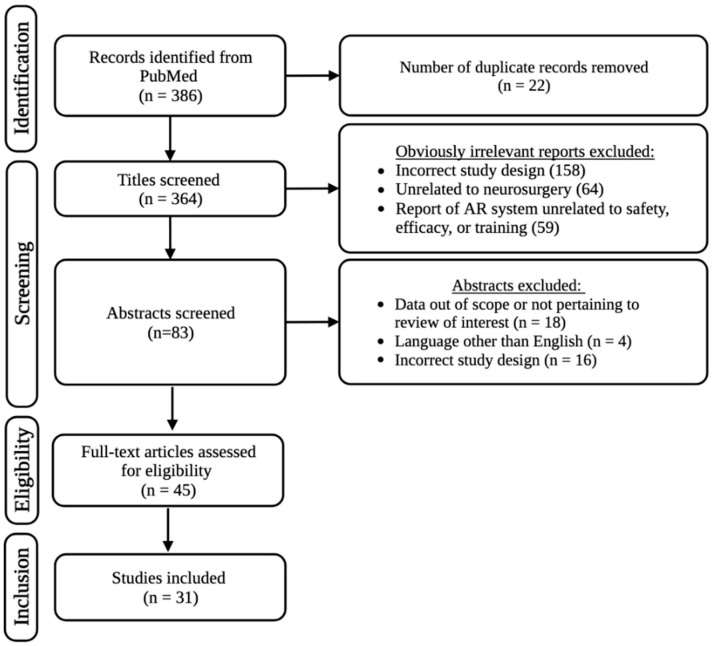
Article selection process as outlined by PRISMA.

**Table 1 medicina-59-01721-t001:** Overview pathology, surgical procedure, anatomic location, AR system, outcome, and application of included studies.

Source	Pathology or Surgical Procedure	Description of AR System	Outcome
**Intraoperative Assistance for Cranial Neurosurgery**			
Almeida et al. [13]	Various cerebral targets	AR created with Swift combined CT images with real-time scans of the skull intraoperatively. AR-identified structures were projected onto a mobile device.	Surgical target recognition with a mean target error of 2.6 ± 1.6 mm.
Gurses et al. [15]	Various white-matter tracts within the brain	Step-by-step dissection of cadaveric brains was used to created 2D and 3D AR models of white-matter tracts within the brain.	2D and 3D models were successfully obtained. Models could be freely rotated in various planes allowing for complete visualization of white-matter tracts.
Finger et al. [16]	Right parietal lesion	Scopus NovaPlan navigation software combined with preoperative CT images were injected and registered onto the preoperative microscope to guide surgical trajectory.	Significant reduction in lesion volume of 47%; 1.2 ± 0.4 mm deviation from target.
Li et al. [18]	Right parietal AVM with evidence of rupture	Microresection of AVM was conducted with an AR holographic projection of angiographic architecture projected onto the cortical surface.	Patient tolerated procedure well and remained stable 6 months after surgery.
Cabrilo et al. [19]	Unruptured aneurysm	Preoperative 3D MRI, CT, and angiography images were used to create virtual segments of individual patient angiographic architecture. These images were injected into the operative microscope.	Continuous monitoring of surgical accuracy was achieved with optimal clipping and minimized exposition.
Rychen et al. [20]	Superficial temporal artery to middle cerebral artery extracranial-intracranial (EC-IC) bypass revascularization surgery	Preoperative 3D images of the STA, middle meningeal artery, and primary motor cortex were segmented and injected into the surgical microscope. These images were projected onto the surgical field intraoperatively.	AR was found to improve neuronavigation.
Sun et al. [21]	Glioma	Preoperative MRI images were obtained and projected onto the surgical microscope intraoperatively.	Tumor resection rate was significantly increased in patients receiving AR assisted surgery (*p* < 0.01).
Ivan et al. [22]	Elective craniotomy for various cerebral tumors	Circumferential tumor border tracing and incision planning was conducted with HoloLens AR glasses using preoperative MRI images.	Five procedures were rated as having an excellent correspondence degree, five had an adequate correspondence degree, and one had poor correspondence.
Bopp et al. [25]	Pituitary lesions requiring transsphenoidal surgery	Preoperative CT images were injected onto the heads-up display intraoperatively.	Use of AR significantly improved surgical accuracy.
Goto et al. [26]	Sellar and parasellar tumors requiring transsphenoidal surgery	Preoperative radiographic data was superimposed onto the surgical field intraoperatively.	The AR navigation system was found to improve patient outcomes in a majority of patients.
**AR as a Training Tool for Cranial Neurosurgery**			
Dominguez-Valasco et al. [27]	Ventricular puncture on a skull phantom	AR was designed to project 3D images of human anatomy on a 3D model of the head.	Enhanced accuracy was achieved for 48 neurosurgeons using AR assistance.
Skryman et al. [28]	Cranial biopsy and EVD insertion on a 3D-printed skull phantom	Intraoperative CT images were combined with integrated video tracking to create an AR system to guide cranial biopsies and EVD insertion.	Mean needle biopsy accuracy was 0.8 mm ± 0.43 mm. Median needle device insertion time was 149 s. Mean EVD insertion accuracy was 2.9 mm ± 0.8 mm. Median insertion time of EVD was 188 s.
Chiou et al. [29]	EVD insertion	Surgical target position, scalpel entry point, and scalpel direction was displayed on a tablet and superimposed on a phantom patient’s head.	High visual acuity was achieved with AR assistance.
Montemurro et al. [30]	En plaque cranial vault meningioma located in the left frontotemporal region	Preoperative MRI images were obtained to generate a head-mounted AR system projected onto a patient-specific mannequin before surgical intervention.	Bone flap tracing ang surgical trajectory planning was achieved with an error of less than ±1 mm.
Van Gestel et al. [31]	Intracranial tumor resection	HoloLens AR was used to determine tumor borders for optimal surgical planning.	registration errors remained below 2.0 mm. tumor delineation was deemed superior in 65% of cases. Surgical planning time was significantly reduced overall.
Ruggiero et al. [32]	Pediatric craniofacial surgery	HoloLens AR was used to determine tumor borders for optimal surgical planning.	Surgical trajectory was successfully traced with an accuracy level of ±1.5 mm.
**Intraoperative Assistance for Spinal Neurosurgery**			
Carl et al. [34]	Intradural tumor resection	AR system was designed from preoperative images and injected into the surgical microscope and on a heads-up display.	AR was successfully applied for all cases and visualization of structures was significantly improved with AR.
Liu et al. [36]	Pedicle screw placement in the thoracic, lumbar, and sacral spine	Three-dimensional images obtained from the O- and C-arm intraoperatively were combined to create a head-mounted AR system.	AR assistance provided an overall surgical accuracy of 98.0%.
Terander et al. [37]	Pedicle screw placement in the thoracic, lumbar, and sacral spine	Three-dimensional images obtained from 4 cameras in the C-arm intraoperatively were combined to create a head-mounted AR system.	AR assistance provided an overall surgical accuracy of 94.1%.
Butler et al. [38]	Degenerative spinal pathology, tumor, and spinal deformity	AR system featured a wireless headset with a transparent near-eye display that projected intraoperative 3D images onto the retina of the surgeon to create a “see-through” surgical field.	Pedicle screw placement time was significantly reduced with AR assistance. Screw placement accuracy was significantly improved with AR assistance.
Kosterhon et al. [39]	Osteotomy in the thoracic and lumbar spine	Three-dimensional virtual resection places were created from preoperative CT images and injected into the surgical microscope.	Patient tolerated the procedure well and experienced no complications.
Hu et al. [41]	Percutaneous vertebroplasty	Three-dimensional images from preoperative CT scans were superimposed onto the surgical field.	Operative time (*p* < 0.001) and surgical accuracy (*p* = 0.028) were significantly improved with AR assistance.
Abe et al. [42]	Percutaneous kyphoplasty in the thoracic, lumbar, and sacral spine	Three-dimensional images from preoperative CT scans were superimposed onto the surgical field via a head-mounted display.	Improved needle projection planning was seen in forty phantom trials and five patients with no significant complications.
Carl et al. [44]	Intradural spinal tumors in the cervical, thoracic, lumbar, and sacral spine	AR system was designed from preoperative CT and MRI images and injected into the surgical microscope and on a heads-up display.	Visualization of anatomical structures was significantly improved with AR. Operation time was significantly reduced with AR assistance.
Trigchelaar et al. [45]	En bloc resection of spinal column lesions	AR system was designed to display augmented images of spinal anatomy within the operative microscope.	Intraoperative use of this AR system resulted in enhanced neuronavigation, smaller surgical incisions, and reductions in bony resection extent with no known patient complications.
Molina et al. [46]	En bloc lumbar spondylectomy osteotomy of chordoma	Three-dimensional images from preoperative CT scans were superimposed onto the surgical field via a head-mounted display.	Patient tolerated procedure well with no postoperative complications.
Sommer et al. [48]	Surgical resection of benign intradural extramedullary tumors	Three-dimensional images from preoperative CT and MRI scans were fused and superimposed onto the surgical field.	This AR system was able to identify tumor location and margins. No postoperative complications were observed.
**AR as a Training Tool for Spinal Neurosurgery**			
Wanivenhaus et al. [50]	Rod manipulation of the cervical, thoracic, lumbar, and sacral spine	HoloLens AR was used to determine spine morphology to guide rod manipulation in a phantom patient.	Rod manipulation was feasible in a phantom patient.
Atzigen et al. [51]	Rod manipulation of the cervical, thoracic, lumbar, and sacral spine	HoloLens AR was used to determine spine morphology to guide rod manipulation in human cadavers.	Rod manipulation was feasible in a human cadaver and operation time was reduced with AR assistance.
Deib et al. [54]	Vertebroplasty, kyphoplasty, and percutaneous discectomy	HoloLens AR was used to project spinal anatomy on a phantom patient.	Intraoperative navigation was improved, and operation times were reduced in trainees using HoloLens AR.
Agten et al. [55]	Lumbar facet joint injections	HoloLens AR was used to project CT images of spinal anatomy on a phantom patient.	HoloLens AR developed from CT images of the spine has an injection accuracy of 97.5% compared to injections guided with CT images alone, which demonstrated an accuracy of 100% in a spine phantom.

**Table 2 medicina-59-01721-t002:** Emerging AR Studies in Spinal and Cranial Neurosurgery.

Study	Study Type	Title	Description
NCT04610411 Recruiting Zurich, Switzerland	Interventional	Evaluation of the Accuracy of Surgical Navigation in Spine Instrumentation Using Augmented Reality	Commercially available AR glasses are used during pedicle screws and rod implants to investigate the accuracy of implant navigation.
NCT03921385 Closed Philadelphia, Pennsylvania, USA	Observational	Augmented Reality Enhanced Surgery: Proof-of-concept Study for the Use of Holographic Technology in Cranial and Spinal Surgery	Studying the feasibility of integrating holographic technology as a heads-up display for navigated spinal and cranial neurosurgical procedures.

Table generated using information from clinicaltrials.gov.

## Data Availability

Not applicable.

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
