# Peer review of "Augmented Reality in Neurosurgery: A New Paradigm for Training"

_medicina, 2023, doi:10.3390/medicina59101721_

Round 1

Reviewer 1 Report (New Reviewer)

The authors present a review of the literature on the use of AR in neurosurgery.

This is interesting work and a useful topic however there are numerous flaws in the work:

- the abstract poorly identifies the type of work the results obtained and the highlights of the conclusions reminding much more of an introduction than an abstract

- the flow-chart of the prism criteria is missing. HOW come no papers before 2013 are included? Specify in the text.

- identifying in the discussion "cranial neurosurgery" without distinguishing the methods between vascular and oncological surgery is not appropriate each of the methods has peculiarities

- It seems that for tumor resection the utility is mainly the trajectory to get to the lesion: are there utilities for studying peritumoral vessels, the vascular pedicle in the case of metastases, or dentification of the peritumoral brain zone in gliomas? 

In general it seems that the main points of technological novelty have not been considered, making the paper flaws at present.

Major revision

Author Response

Response to Reviewer 1 Comments

Point 1 explains that the abstract poorly identifies the type of work the results obtained and the highlights of the conclusions reminding much more of an introduction than an abstract.

Response 1: To improve this aspect of our review, we have revised the abstract to include more information about the types of studies, specifically original research, and case reports. We have additionally highlighted specific findings from our review, most notably the most common applications of AR for intraoperative use and training. Our abstract now reads:

“Augmented reality (AR) involves the overlay of computer-generated images onto the user’s real-world visual field to modify or enhance the user’s visual experience. With respect to neurosurgery, AR integrates preoperative and intraoperative imaging data to create an enriched surgical experience that has been shown to improve surgical planning, refine neuronavigation, and reduce operation time. In addition, AR has the potential to serve as a valuable training tool for neurosurgeons in a way that minimizes patient risk while facilitating comprehensive training opportunities. The increased use of AR in neurosurgery over the past decade has led to innovative research endeavors which aim to develop novel, more efficient AR systems while also improving and refining present ones. In this review, we provide a concise overview of AR, detail current and emerging uses of AR in neurosurgery and neurosurgical training, discuss the limitations of AR, and provide future research directions. Following the guidelines of the Preferred Reporting Items for Systematic Reviews and Meta-Analyses (PRISMA), 386 articles were initially identified. Two independent reviewers (GH and AC) assessed article eligibility for inclusion and 45 articles are included in this review. The literature search included original (retrospective and prospective) articles and case reports published in English between 2013 and 2023. AR-assistance has shown promise within neuro-oncology, spinal neurosurgery, neurovascular surgery, skull base surgery, and pediatric neurosurgery. Intraoperative use of AR was found to primarily assist with surgical planning and neuronavigation. Similarly, AR assistance for neurosurgical training focused primarily on surgical planning and neuronavigation. However, studies included in this review utilize small sample sizes and remain largely in the preliminary phase. Thus, future research must be conducted to further refine AR systems before widespread intraoperative and educational use.”

Point 2: The flow-chart of the prism criteria is missing. HOW come no papers before 2013 are included? Specify in the text.

Response 2: We have updated our manuscript to include a PRISMA chart. To clarify why no papers before 2013 were included, we have included an additional sentence in section 3 which states “The search was limited to studies published between 2013 and 2023 in attempts to provide the most updated information which we defined as published within the past 10 years.”

Point 3 stated “Identifying in the discussion "cranial neurosurgery" without distinguishing the methods between vascular and oncological surgery is not appropriate each of the methods has peculiarities.”

Response 3: The term “cranial neurpsurgery” is not included in section 6. Conclusion and Future Research Directions. Within section 4.1 Current use of AR in the Operating Room for Cranial Neurosurgery, there are separate paragraphs for both vascular and oncological surgery explaining the intricacies and individual differences between both types of surgery.   

Point 4 stated it seems the utility for tumor resection lies mainly in the trajectory to get to the lesion and asks if there are utilities for studying peritumoral vessels, the vascular pedicle in the case of metastases, or dentification of the peritumoral brain zone in gliomas.

Response 4: At this point in time, there is a gap in the literature regarding the use of AR for studying peritumoral vessels, the vascular pedicle in the case of metastases, or dentification of the peritumoral brain zone in gliomas. However, we agree that this is an important aspect of AR use that should be addressed. Thus, we have included a need for future research targeting these aspects of care in section 4.2 Current uses of AR as a Training Tool for Cranial Neurosurgery. The sentence reads “It is important to note that the intraoperative use of AR for neuro-oncological surgery primarily focuses on surgical planning and intraoperative neuronavigation. However, there are other important aspects of cranial malignancy pathology such as peritumor vessels, vascular pedicle dysregulation, and dentification of the peritumoral brain zone where the use of AR for treatment has not been explored in the literature. Thus, future studies should be conducted to investigate how AR can be used to study peritumoral vessels, the vascular pedicle in the case of metastases, and dentification of the peritumoral brain zone to ultimately expand the use of AR for intraoperative treatment of various malignancies.”

Reviewer 2 Report (New Reviewer)

This systematic review was performed on 45 articles to determine the augmented reality in neurosurgery.

The following points seem worthy to mention:

1-      The PRISMA diagram must be prepared in the method section.

2-      A table containing the characteristics of the 45 included articles and multi-parametric assessment for AR systems in neurosurgery should be prepared in the results section for example: author, date, pathology, location, real data source, virtual data source, tracking modality, registration technique, visualization processing, display type, perception location.

3-      A table containing the pathology of neurosurgical lesions treated with the aid of augmented reality should be prepared in the results section.

Please  recheck after reision of the article

Author Response

Response to Reviewer 2 Comments

Point 1 indicated to include a PRISMA diagram in the methods section.

Response 1: We have included a PRISMA diagram in the methods section (Figure 2)

Point 2 indicated a table containing the characteristics of the articles must be included. Point 3 explains a table with the pathology of lesions must also be included.

Response 2/3: To address comments 2 and 3 together, we have updated our manuscript to include a table that outlines all of the studies included in this review. See table 1 in the revised manuscript for specifics.

This manuscript is a resubmission of an earlier submission. The following is a list of the peer review reports and author responses from that submission.

Round 1

Reviewer 1 Report

The authors reported the state of the art of Augmented Reality in training neurosurgery. The paper is just a narrative/description review and not a systematic review ( not report any type of PRISMA METHODS).

After careful review, I suggest the following items:

- English needs a review;

- any abbreviations need a full form;

- some imagines/flowcharts could improve the impact of paper;

- a table about the studies recruited Could be helpful.

Need a revision.  

Author Response

Point 1: Suggested to review the paper and review our use of English.

Response 1: In response to reviewer 1’s suggestion, we have revised various elements of the manuscript to improve our use of English. Throughout the manuscript, sentences have been revised for clarity and simplicity.

Point 2: Notes all abbreviations need a full form.

Response 2: At the end of our manuscript before the references section is a list of all abbreviations with their included full forms. We have additionally checked to ensure all abbreviations have a full form listed.

Point 3: Suggested to include images/flowcharts to improve the overall impact and quality of the paper.

Response 3: We agree with reviewer 1 that the incorporation of a visual figure would improve the overall impact and quality of our manuscript. As such, we have included figure 1 into our manuscript which shows a brief overview of how AR is currently used in neurosurgery.

Point 4: Suggests including a table about recruited studies in the manuscript.

Response 4: We have included Table 1 “Emerging AR Studies in Cranial and Spinal Neurosurgery” in the original version of our manuscript to highlight these studies. The table contains few references as there are few ongoing clinical trials regarding the use of AR in neurosurgery.

Reviewer 2 Report

Thank you for this opportunity to review the paper.

The authors cover the main points of how AR is being used in neurosurgery, both in surgery and in training inc. discussing the pros and cons of using AR in neurosurgery, and discussing some of the future directions of research in this area.

I think one thing that could add to the review is a discussion of the cost of AR systems. It is an important consideration, as AR systems can be quite expensive. Also discussing the availability of AR systems, as this is another important factor that will affect the widespread adoption of AR in neurosurgery.

Overall, I think this is a well-written and informative review of AR in neurosurgery, summarising the current state of the field, and the paper has highlighted some of the potential benefits and challenges of using AR in this setting.

Author Response

We thank reviewer 2 for their positive feedback and kind words regarding our manuscript. We have made the following revisions:

Comment 1: Suggested to include information in the manuscript about both the cost and availability of AR systems.

Response 1: We agree with reviewer 2 that our manuscript could benefit from a more through description of the cost and availability of AR systems. As such, we have included a more detailed description in our conclusion. The paragraph we have elaborated on now reads:

“The overall cost of AR technology can additionally serve as a barrier to clinical implementation, especially in smaller hospitals or clinics that may not benefit from sig-nificant research grants or large private donors. Adequate funding is necessary to maintain AR systems including repairs, software development, hardware maintenance, and training. AR systems are constantly evolving, and significant monetary input is necessary to keep up with software and hardware advances. The use of AR in the in-traoperative environment requires the presence of additional staff such as engineers or software developers to attend to technological issues that may arise during surgery. Without the presence of these staff members, surgical times can be delayed, AR systems can experience significant damage, and patient safety can be compromised. It should not be surprising to note that the cost of AR systems ultimately impacts the availability of AR systems. The use of AR in neurosurgery remains largely in the preclinical phases, highlighting the need for continued research efforts before widespread clinical adoption. Many institutions currently utilizing AR in neurosurgery are conducting laborato-ry-based experiments and human clinical trials. Thus, institutions must possess sufficient funding to support these research efforts to ultimately refine AR systems before clinical implementation. Furthermore, if institutions lack the necessary funding to support the use of AR during surgery, it is possible that the AR software or hardware could become faulty, resulting in a significant financial loss. More importantly, use of damaged AR systems or AR systems that are not properly maintained can pose a significant risk to patient safety. As such, if institutions cannot afford to properly maintain AR systems and keep up with the advances in technology, it is impossible to utilize such systems during surgery without creating a significant risk of financial loss, equipment damage, patient morbidity, and patient mortality which limits the overall availability of AR systems. This is a challenge that must be addressed as AR continues to develop and evolve to ensure all patients have access to the highest quality of neurosurgical care.”

Round 2

Reviewer 1 Report

I appreciate the effort to improve the paper. 
They added a figure and tagli to improve the impact of the paper. 
Good job

Good english